# Guanidines Conjugated with Cell-Penetrating Peptides: A New Approach for the Development of Antileishmanial Molecules

**DOI:** 10.3390/molecules30020264

**Published:** 2025-01-10

**Authors:** João Victor Marcelino de Souza, Natalia C. S. Costa, Maria C. O. Arruda Brasil, Luana Ribeiro dos Anjos, Renata Priscila Barros de Menezes, Eduardo Henrique Zampieri, Jhonatan Santos de Lima, Angela Maria Arenas Velasquez, Luciana Scotti, Marcus Tullius Scotti, Marcia A. S. Graminha, Eduardo R. Pérez Gonzalez, Eduardo Maffud Cilli

**Affiliations:** 1Department of Biochemistry and Organic Chemistry, Institute of Chemistry, São Paulo State University (UNESP), Araraquara 14800-060, SP, Brazil; 2School of Pharmaceutical Sciences, São Paulo State University (UNESP), Araraquara 14800-903, SP, Brazil; natalia.costa@unesp.br (N.C.S.C.);; 3Fine Organic Chemistry Lab, School of Sciences and Technology, São Paulo State University (UNESP), Presidente Prudente 19060-080, SP, Brazil; luana.anjos@unesp.br (L.R.d.A.);; 4Natural Products and Synthetic Bioactives Postgraduation Program, Federal Paraiba University (UFPB), João Pessoa 58051-900, PB, Brazil

**Keywords:** cell penetration peptide, guanidine, bioconjugate, Leishmania, cysteine protease, selectivity

## Abstract

Leishmaniasis is a neglected tropical disease caused by a protozoan of the genus Leishmania, which has visceral and cutaneous forms. The symptoms of leishmaniasis include high fever and weakness, and the cutaneous infection also causes lesions under the skin. The drugs used to treat leishmaniasis have become less effective due to the resistance mechanisms of the protozoa. In addition, the current compounds have low selectivity for the pathogen, leading to various side effects, which results in lower adherence to treatment. Various strategies were developed to solve this problem. The bioconjugation between natural compounds with antimicrobial activity and cell-penetrating peptides could alleviate the resistance and toxicity of current treatments. This work aims to conjugate the cell penetration peptide TAT to the guanidine GVL1. The GVL1-TAT bioconjugate exhibited leishmanicidal activity against *Leishmania amazonensis* and *Leishmania infantum* with a high selectivity index. In addition, the bioconjugate was more active against the intracellular enzyme CPP than the individual compounds. This target is very important for the viability and virulence of the parasite within the host cell. Docking studies confirmed the higher interaction of the conjugate with CPP and suggested that other proteins, such as trypanothione reductase, could be targeted. Thus, the data indicated that guanidines conjugated with cell-penetrating peptides could be a good approach for developing antileishmanial molecules.

## 1. Introduction

Leishmaniases are neglected tropical and subtropical diseases. The intracellular parasite responsible for infecting macrophages is the protozoan of the genus *Leishmania* spp. of the *Trypanosomatidae* family, whose vector is the female *phlebotomine* insect (*Phlebotominae*) [1,2].

The disease can take different forms, the main ones being visceral and cutaneous (tegumentary). Visceral leishmaniasis presents symptoms such as high fever, dry cough, diarrhea, weakness, and visceromegaly. As the disease progresses, the patient suffers weight loss and weakening, increasing susceptibility to infections by opportunistic pathogens, as well as the possibility of hemorrhagic conditions. These symptoms can lead to death if left untreated. Tegumentary leishmaniasis causes painless sores in the region of the insect bite, with eruptions, which can progress to the mucocutaneous form when it is found in the nose, oropharynx, palates, lips, and tongue, in which destructive lesions appear in the mucosa [3].

The first-line treatments for leishmaniasis are pentavalent antimonials, and the second choice is Amphotericin B. These compounds have side effects, liver and kidney toxicity, and reduced action, which, due to their prolonged use, result in parasite resistance [4]. Thus, the compromised efficacy of the compounds currently indicates the need to develop new treatments [5] and the search for new therapeutic molecules.

Antimicrobial peptides (AMPs) are a class of compounds that can act against bacteria, fungi, protozoa, and viruses. These molecules are made of a relatively short chain of amino acids with a low molecular weight compared to proteins. AMPs can exert direct activity on the membrane to damage its structure or act on specific targets, such as intracellular enzymes and the immune system [6]. In addition to peptides, which show activity toward different pathogens, bioconjugates can be developed by linking an AMP to other organic molecules [7].

Guanidine compounds were investigated as potential treatments against parasite diseases [8,9,10], with promising results in the development of molecules for treating leishmaniasis [11]. Guanidines are naturally found in plants or microorganisms and have various biological activities [12]. One possible mechanism of action of guanidines on the parasite is their ability to inhibit enzymes called cysteine proteases (CPs), which are essential for the parasite’s survival. They can also selectively induce oxidative stress in the parasite, leading to its death [13]. In *Leishmania*, these proteases are essential in processes including differentiation, nutrition, host cell infection, and evasion of the host’s immune response [14]. The literature shows that the knockout of the CPB gene in *L. mexicana* could impair the infectivity of the parasites against macrophages, which leads to a delay in the progression of the disease in vivo. In addition to silencing this gene, it interfered with several virulence factors of the parasite [15]. Tests with different types of guanidines have demonstrated that they can inhibit up to 73% of the activity of these enzymes [13]. However, the molecule has low solubility, which is an important limiting factor because this property directly impacts the absorption, bioavailability, and speed of action of drugs.

Bioconjugates can confer greater selectivity, solubility, lower cytotoxicity, and consequently better therapeutic activity. The union of peptides with guanidines forms more active molecules since peptides favor the selectivity and entry of guanidines through the parasite’s cell membrane, facilitating the interaction of guanidines and the peptides with intracellular targets. Initially, our group tested guanidines conjugated to AMPs called TSHa, with good results and greater efficacy than the guanidine and peptide alone [16]. The bioconjugate was more potent and exhibited higher selectivity than the individual compounds. The improved activity could be due to an increase in the concentration of intracellular guanidine through the formation of transient pores.

Cell penetration peptides (CPPs) are small (5–30 amino acids) and can transport bioactive compounds into the cells. These peptides are cationic, amphipathic, anionic, and hydrophobic [17]. Cationic CPPs mainly consist of many residues of arginine and lysine. In this work, this class of peptides was used to increase the solubility and internalize guanidine. The transactivator of transcription peptide (TAT) [18,19] was studied to diagnose and treat various diseases, including parasitic ones [20]. In addition, TAT was shown to accumulate inside the promastigote form of *L. donovani* [21] and in macrophages [22], indicating that this peptide is interesting for internalizing bioactive compounds.

This work aims to develop bioconjugates containing the guanidine (Z)-4-((2-benzoyl-3-(4-bromophenyl)guanidino)methyl) benzoic acid, also known as GVL1 [16], and the CPP known as TAT(47–56). This conjugation aims to potentiate the leishmanicidal effects of guanidine molecules and confirm that the transport of guanidines into the cell is the reason for the increased potency of the bioconjugate previously studied.

## 2. Results and Discussion

### 2.1. GVL1-TAT Synthesis and Characterization

The GVL1-TAT bioconjugate was synthesized using solid-phase peptide synthesis (SPPS) (Figure 1). The protocol was straightforward, requiring no additional procedures beyond those used in SPPS. The GVL1 guanidine was coupled to the TAT by a linkage to the N-terminal group of the peptide. The coupling of the GVL1 guanidine was more efficient using HATU/DIEA than the previously used PyBOP/DIEA [16].

The peptide was purified by reversed-phase HPLC in semi-preparative mode. The material obtained was analyzed by mass spectrometry, confirming that the desired material was obtained (Figure 2).

### 2.2. Biological Assays

#### 2.2.1. Evaluation of Leishmanicidal and Cytotoxic Activities

One focus of this work is the development of molecules against cutaneous leishmaniasis (CL), caused by several species of *Leishmania*, requiring specific treatment for each species. Globally, CL affects 12 million people, and 2 million new cases occur annually [3,23,24].

The first therapeutic options available are intralesional; however, in cases of mucocutaneous *Leishmania* or disseminated cutaneous leishmaniasis, a systemic therapeutic approach is employed. Therefore, the development of less toxic and less painful therapies is needed, especially for children [20,23]. However, the financial incentives are limited for developing new molecules, vaccines, and diagnostics for its treatment because CL mainly affects low-income people [5,25,26,27].

*L. amazonensis* is one of the species that causes CL. The leishmanicidal assays against *L. amazonensis* showed that the TAT peptide alone had no activity (Table 1), while GVL1 had an IC_50_ of 51.8 and 7.5 μM for the promastigote and amastigote forms, respectively. The GVL1-TAT bioconjugate exhibited the highest activity in both the promastigote and amastigote forms, with an IC_50_ of 5.3 and 0.8 μM, respectively. In amastigotes, the increase in activity was approximately 10 times greater for the bioconjugate than guanidine alone. Thus, the transport of GVL1 into the cell or parasite is important for its activity.

This study also proved that guanidine has an intracellular mechanism of action, and facilitating its passage through the membrane increases its activity. In addition, this compound did not show toxicity against macrophages at the highest concentration studied. Compared to Amphotericin B, the bioconjugate GVL1-TAT had almost 2 times lower activity in the promastigote form and similar results in the amastigote form. Nonetheless, it was less toxic than the drug, with a higher selectivity index (SI) for GVL1-TAT (1250) than Amp B (131). This index is calculated as the ratio between the IC_50_ and CC_50_ values (Table 1).

The results were like those obtained by GVL1-TSHa [16], with IC_50_ values of 5.4 and 0.33 μM for the promastigote and amastigote forms of *L. amazonensis*, respectively. The TSHa peptide is an antimicrobial peptide, and its action mechanism was described as pore formation [28,29]. These data indicated that pore formation could facilitate the deeper entrance of the guanidine and promote its inhibitory activity in inner proteins.

The use of CPP against parasites is still minimally explored. Silva et al. 2020 evaluated the antileishmanial activity of combined therapy between crotamine and the pentavalent antimonial Glucantime [30]. They showed that this pharmacological association improved the inhibition of the amastigotes of *L. amazonensis*. Illa et al. 2020 [21] found that new hybrid CPP peptides formed by the repetition of a dipeptide unit (cyclobutane amino acid and either a cis- or trans-γ-amino-l-proline) were microbicidal on the protozoan parasite *Leishmania* beyond 25 μM [21]. In addition, the doxorubicin (Dox)-TAT(48-57) toxicity against *L. donovani* occurred in concentrations above 10 μM. Tat(48-60) was not toxic in *L. donovani*, but the peptide-linked fluorescent miltefosine derivatives Quasar 670-Lys(Cys-S-BDP-MT)Tat and Ac-Lys(COCH_2_-S-BDP-MT)Tat were active in proliferation of the *L. donovani* promastigotes at 7.5 μM [22]. The comparison of our data with those obtained for the bioconjugate showed that GVL1-TAT is more active than the other molecules previously described.

Parasitic infections, such as leishmaniasis, use several strategies to establish the infection with the host’s immune system turned off so that no immune response is generated against the host cells and the parasite can establish itself and proliferate. To do this, the parasite uses the formation of vacuoles that contain the promastigote form after being inside, which will be converted into amastigote [31]. In these processes, the replicated amastigotes are transmitted to healthy cells, leading to the amplification of the infection [32], which can be assessed by the infection index (percentage of infected cells × number of parasites per infected cell), which evaluates this transmission process. In this work, the infection index was determined using *L. amazonensis* for 24 h (Figure 3) of treatment.

For bioconjugate GVL1-TAT, after 24 h, the infection rate decreased 14.2-fold compared to the infected cells that were not treated. In comparison to the control drug Amp B, the bioconjugate was 7.6 times more effective in decreasing the infection rate, while the TAT peptide and GVL1 showed similar values on the infection index. The better reduction found for GVL1-TAT may be explained by the higher capacity of this molecule to cross epithelial barriers, serving as a carrier to deliver GVL1 to its target [22,33]. The bioconjugation strategy overcomes some challenges in the treatment of infections with intracellular pathogens, such as *Leishmania*, since the parasite has mechanisms to evade the host’s phagocytic killing mechanisms. With the use of specific treatments, the increase in multidrug resistance phenotypes in pathogens can be avoided [31].

The bioconjugation strategy was also evaluated against *L. infantum*, which causes visceral leishmaniasis. This is the most severe form of the disease, characterized by fever and weight loss, affecting internal organs, including the spleen and liver, and the mortality rate is up to 95% if left untreated [34]. There is no universal treatment for visceral leishmaniasis, and guidelines vary between regions, the causative parasite, the patient’s immunological status, and the local availability of therapy. Therefore, the search for new molecules for this form of disease is essential.

The leishmanicidal assays against *L. infantum* showed that the TAT peptide alone had no leishmanicidal activity (Table 2), while the GVL1-TAT bioconjugate presented activity against the promastigote and amastigote forms, with an IC_50_ of 1.2 and 3.8 μM, respectively. In amastigotes, the increase in activity was approximately 3 times greater for the bioconjugate than guanidine alone. The peptide–drug conjugates (PDCs), including GVL1-TAT, play an essential role in the ability of the molecules to penetrate the membrane, reach specific targets inside the cell, and increase the effective intracellular concentration of a drug–CPP conjugate [35]. Compared to Amphotericin B, bioconjugate GVL1-TAT had almost 5 times lower activity in the promastigote form. Nonetheless, it was less toxic than the drug, i.e., it had a higher SI (526) than Amp B (19) (Table 2).

Regarding the ability to infect new macrophages, bioconjugate GVL1-TAT, after 24 h, decreased the infection rate 11.2-fold at 10 μM (Figure 4) compared to infected cells that were not treated. In comparison to the control drug Amp B, the bioconjugate was 3.5 times more effective, while the TAT peptide and GVL1 had smaller and similar infection index values. In addition, the decrease in the infection rate for *L. infantum* was dose-dependent, that is, the higher the dose tested, the greater its ability to prevent the infection of new macrophages.

#### 2.2.2. Stability Assay with Active Fetal Bovine Serum

Peptide candidate degradation by plasma proteases and hepatic and renal enzymes has hindered the transition of peptides from the laboratory to the market [36]. Several strategies were employed to increase the stability of peptides, including the incorporation of non-natural amino acids, modifications of the N- and/or C-terminal, cyclization, the use of non-peptide structures (peptidomimetics), and multimerization [37,38,39]. Therefore, stability studies were carried out with TAT, GVL1, Amp B, and GVL1-TAT in the presence of the *Leishmania* parasite in its promastigote form, with active fetal bovine serum added to the culture medium. This serum contains proteases that could degrade these molecules.

As observed previously, TAT did not display anti-promastigote activity against the species tested. The GVL1 compound maintained its biological activity against both species. The bioconjugate GVL1-TAT showed an IC_50_ of 2.6 μM (Table 3) for the promastigote form of *L. amazonensis* and an IC_50_ of 1.2 μM for *L. infantum*, demonstrating significant biological activity. Hence, although the compounds showed less activity and some type of degradation in the presence of an active serum, they nonetheless maintained relevant biological activity.

### 2.3. Vesicle Permeabilization

To verify whether the target of the bioconjugates was the cell membrane, permeabilization studies of vesicles labeled with carboxyfluorescein were carried out. The data obtained showed that none of the compounds evaluated, whether guanidines, TAT peptides, or bioconjugates, could permeabilize vesicle mimetics of parasite cells (Figure 5), indicating that the mechanism of action does not involve the cell membrane. As illustrated in Figure 5 and Figure 6, the analyte in question (peptide, bioconjugate, or guanidine) was added within 120 s, and after an interval of 360 s, the behavior of the membrane mimetic was reanalyzed. After this time, 10% triton was added so that the vesicles could completely release the carboxyfluorescein. The data showed that the compounds did not affect the membrane, as can be seen in Figure 5, strengthening the intracellular target hypothesis.

For the tests carried out on eukaryotic cell mimetics, GVL1-TAT displays a subtle curve in Figure 6. One hypothesis for this phenomenon is the formation of transient pores [40], which may be a mechanism for the entry of guanidine and the bioconjugate inside the cell. However, no damage was observed in the vesicles, which only occurred after the addition of the detergent, indicating that the bioconjugate exhibits no toxicity even at high concentrations.

### 2.4. Cysteine Protease Enzyme Inhibition

In addition to the membrane, peptides may have other therapeutic targets in *Leishmania*. Enzymes are essential for functional integrity and virulence, highlighting the fundamental role of cysteine proteases (CPB) in the infectious process and their inhibition to have a high therapeutic potential, enabling the design of new drugs [41,42,43]. In *L. mexicana*, three cysteine protease genes were identified as virulence factors: lmcpa, lmcpb, and lmcpc. Of these, lmcpb encodes a cathepsin L-like protein and is the most abundant in the amastigote form of the parasite, making this enzyme an important therapeutic target. Guanidinic compounds can also target the cysteine protease, highlighting the compound E64 (L-trans-epoxysuccinyl-leucylamido-(4-guanidino)butane), which, in the nanomolar range, is a potent irreversible inhibitor of *Leishmania* CPB [13]. Compounds belonging to the azapeptide class were reported to inhibit cysteine proteases such as papain, cathepsin B, and cathepsin K. One of these compounds, Z-Arg-Leu-Val-Gly-Ile-Val-Ome, is a potent inhibitor of cathepsin B, with a Ki value of 0.088 nmol L^−1^ [44], demonstrating the high potential of peptides as cysteine protease inhibitors.

In addition, guanidine LQOF-G6, a precursor of GVL1, showed selective inhibitory activity on *Leishmania major* cysteine protease LmCPB2.8∆CTE (CPB) with an IC_50_ of 6.0 μM [13]. We evaluated the action of GVL1, TAT, and GVL1-TAT on cysteine protease B (CPB), an important enzyme for the cell viability of the parasite. The bioconjugate exhibited great activity against this enzyme (Table 4), where approximately 92% of enzymatic activity was inhibited. On the other hand, the TAT peptide and isolated guanidine did not show satisfactory activity; thus, the bioconjugation also increased the affinity for the enzyme, which is one of the possible mechanisms of action for the molecule.

Our data on the CPB enzyme inhibition study may also explain the low infection index of the peptide bioconjugates at the concentrations tested against *L. amazonensis* and *L. infantum*. Cysteine proteases may interfere with the parasite infection process. *Leishmania* expresses high levels of several classes of CPs belonging to the papain family, which are crucial for the parasite’s metabolism, reproduction, and intracellular survival [42,43,45]. During the process of *Leishmania* infection in macrophages, CPB modulates the host’s responses by negatively regulating protective Th1 immune responses, particularly IFN-γ production, through the degradation of the transcription factor NF-κB and subsequent inhibition of IL-12 production by infected macrophages. Therefore, the protozoan cysteine protease, CPB, can alter host macrophage signaling by increasing IL-12 transcription, which hampers the establishment of infection [15,44,46,47]. The literature, including the mechanism of action of guanidine compounds, demonstrates the fundamental role of parasite cysteine proteases in the infectious process and suggests the high therapeutic potential of inhibiting these enzymes [41,43,44]. This may justify why GVL1-TAT presented a lower infection rate than GVL1 and the control drug, Amp B. Another important point is that CPB is more expressed in the amastigote form, which plays an important role in the interaction between *Leishmania* and its mammalian host, in which the inhibition of this enzyme can decrease macrophage infectivity.

### 2.5. Docking Studies

To confirm the action mechanism of GVL1-TAT on cysteine protease and in trypanothione reductase (TR), other therapeutic targets against *Leishmania* docking studies were conducted. TR is a NADPH-dependent flavoenzyme that sustains a reduced intracellular environment. These enzymes are constitutively expressed throughout the parasite’s life cycle, making it crucial for the parasite’s survival [48,49].

To perform molecular docking studies, it is essential to have the three-dimensional structures of the target proteins. The three-dimensional structures of the enzymes trypanothione reductase and cysteine protease were not available in the literature or major crystallographic databases such as the Protein Data Bank. Therefore, these protein structures were generated using the AlphaFold web tool. High-resolution X-ray crystallographic data are indispensable for deciphering enzyme mechanisms, understanding the architecture of active sites, and elucidating protein–ligand interactions. These structural insights serve as a cornerstone for precise docking studies and the rational design of potent inhibitors. However, when experimental crystal structures are unavailable, alternative computational approaches, such as homology modeling and advanced tools like AlphaFold, become crucial. Homology modeling employs structurally similar templates to construct reliable models of the target protein. In contrast, AlphaFold represents a groundbreaking tool capable of accurately predicting protein structures de novo, offering a robust solution in the absence of experimental data [50,51,52].

The docking results, expressed in the MolDock score, indicate that the lower the energy level, the greater the predicted interaction stability. Table 5 presents the energy values of the molecules for each of the analyzed proteins.

For the cysteine protease protein, the molecular docking results corroborate the in vitro findings. GVL1-TAT showed a lower interaction energy than the peptide and guanidine alone, reflecting the in vitro results, where its activity increased upon conjugation.

Lasakosvitsch et al. 2003 [53] aligned the amino acid sequences of cysteine proteases from five *Leishmania* species and found that four catalytic site residues—Cys 153, His 289, Gly 151, and Asn 309—are conserved across species. In this context, the interactions of TAT, GVL1, and GVL1-TAT with the cysteine proteases were analyzed to understand how the activity profiles of these molecules relate to their interactions with the active site. The interactions between the molecules and the active site residues provide insight into the observed energy values and in vitro activities, as detailed in Table 6 and Appendix A. TAT and GVL1 were the molecules with the lowest interaction energies and correspondingly low leishmanicidal in vitro activity levels (Table 1 and Table 2) and CPB enzyme inhibition (Table 4). Analyzing their interactions reveals that these molecules do not form hydrogen bonds with the active site residues of cysteine proteases, and their hydrophobic interactions do not involve the catalytic residues of the active site, explaining their lower activity. The GVL1-TAT peptide interacts with all four catalytic residues of the cysteine protease active sites. These include hydrogen bonds with Cys 153 and His 289, and hydrophobic interactions with Gly 151 and Asn 309. These interactions help explain the significant inhibition percentage of this peptide observed in the in vitro study (Table 4).

With the aim of developing multitargeted compounds, the efficacy of a single binding molecule against the second enzyme trypanothione reductase was explored. Multitargeted compounds may act on more than one biological target in interconnected biochemical pathways responsible for the pathophysiology of multifactorial diseases, such as leishmaniasis [54,55].

To evaluate that other enzymes could be inhibited by a conjugate, docking with trypanothione reductase was performed. The results showed similar predicted interaction stability between the molecules. The GVL1 + TAT and GVL1-TAT complexes demonstrated potentially greater stability than the molecules in their unconjugated forms, with higher interactions with bioconjugates. An important form of host defense against the parasite during the establishment of infection is oxidative stress, so that the survival of the parasite depends mainly on the ability to resist this attack. Infectious trypanosomatids lack catalase and other conventional redox control systems and base their defense on trypanothione, an unusual variant of glutathione, the main agent in maintaining thiol homeostasis. This enzyme is considered a good target for drugs because it is essential for the survival of the parasite. It wasvalidated as a target in both *Leishmania* and *Trypanosoma* [56,57]. Thus, the GVL1-TAT bioconjugate appears to have good potential as a therapeutic candidate for leishmaniasis, having dual targets against different species of the parasite.

## 3. Materials and Methods

### 3.1. Peptide Synthesis

The synthesis method is based on the addition of residue by residue to grow the peptide chain, which, in turn, is covalently attached by its C-terminal amino acid to the reactive sites on a solid support (resin) by means of its carboxyl group. Wang resin was employed to obtain the peptide called TAT with a free C-terminal carboxyl. The peptides were synthesized manually, according to the standard FMOC strategy, as previously described [16,58].

The bioconjugation strategy involved coupling the GVL1 compound, which was in 1.5-fold excess over the amino component. An activating reagent other than the one used to synthesize the peptide was then used, namely 2-(1-H-7-azabenzotriazol-1-yl)-1,1,3,3-tetramethyluronium hexafluorophosphate (HATU) in the presence of N,N-diisopropylethylamine (DIPEA), in a reaction that took place over 24 h under heating at 50 °C. The pH also had to be controlled and was kept at approximately 9.

The resulting products were purified by semi-preparative HPLC, employing a C18 reversed-phase column (Jupiter Proteo, 25 cm × 10 mm, 5 μm particles). Elution was performed using water with 0.045% TFA (solvent A) and acetonitrile with 0.036% TFA (solvent B), at a flow rate of 5 mL min^−1^, with detection at 220 nm. The purity of each fraction was determined using HPLC in analytical mode, with a C18 column (25 cm × 10 mm) and elution using a gradient from 5 to 95% solvent B over 30 min, with an eluent flow rate of 1 mL min^−1^.

Detection was performed at 220 nm with a Shimadzu spectrometer (Shimadzu, Tokyo, Japan). The molecular weights of the peptides were analyzed using a mass spectrometer in ESI-IT-MS configuration (LCQ Fleet, Thermo Fisher Scientific, Waltham, MA, USA), with an ion trap analyzer in positive electrospray mode (M+H)^+^ and a range of 200–2000 g mol^−1^.

### 3.2. Biological Assays

*L. amazonensis* promastigotes (MPRO/BR/1972/M1841-LV-79) were grown in LIT culture medium, while *L. infantum* promastigotes (MNYC/BZ/62/M379) were cultivated in 199 culture medium (Sigma-Aldrich, St. Louis, MO, USA) supplemented with 10% heat-inactivated fetal bovine serum (FBS; Gibco/Invitrogen) [59] and in 10% urine, at 28 °C, until reaching the mid-log phase of growth. Macrophages were sourced from the peritoneal cavity of Swiss mice, using the methodology outlined by de Almeida et al. 2017 [60]. For the anti-promastigote assays, *L. infantum* and *L. amazonensis* cultures in the exponential growth phase were transferred to 96-well plates at a final concentration of 1 × 10^7^ promastigotes mL^−1^. Subsequently, the tested compounds were added at concentrations ranging from 1.28 to 100 µmol L^−1^ to assess their anti-promastigote activity. After 72 h, viable promastigotes were determined using the MTT method [61].

Anti-promastigote and anti-amastigote assays, the infection index (percentage of infected cells * number of intracellular parasites/numbers of infected cells), and their cytotoxic activity were conducted as previously described [59].

### 3.3. Serum Stability

Peptide stability was investigated using anti-promastigote assays (as described previously [16]) with *L. amazonensis* and *L. infantum* using LIT or 199 culture media with active fetal bovine serum.

### 3.4. Vesicle Preparation and Permeabilization Assay

Large unilamellar vesicles (LUVs) composed of POPC (1-palmitoyl-2-oleoyl-sn-glycero-3-phosphatidylcholine) and cholesterol (Chol) (4:1) or POPC, POPS (1-palmitoyl-2-oleoyl-sn-glycero-3-serine), and ergosterol (Erg) (8:1:1) were prepared according to the methods described by Lorenzón et al. 2014 [62]. The release rate of CF from the vesicles was measured using the fluorescence intensity at 490 and 512 nm after the addition of peptides in concentrations of 50 μM and 10 μM, respectively.

### 3.5. Cysteine Protease (CPB) Enzyme Inhibition Assay

CPB inhibition experiments were performed as described previously by Moreira et al. 2022 [13]. The results were expressed as mean ± standard deviation of two independent replicates.

### 3.6. Molecular Docking

The three-dimensional structures of the proteins—trypanothione reductase (GR) enzyme and cysteine protease—were obtained using the AlphaFold web tool, which computationally predicts protein structures with high accuracy. The amino acid sequences of each protein are available in the UniProt database [63]: A5JV95 for trypanothione reductase and Q867S4 for cysteine protease.

The active sites of these proteins were identified using the ProteinsPlus platform [64] with the DoGSiteScorer tool [65,66]. This tool employs a Difference of Gaussian (DoG) method to detect and analyze cavities on the protein surfaces, which are potential binding sites for ligands. These cavities are crucial for drug development as they allow the prediction of which sites are most likely to be functional and bind to molecules of interest, such as inhibitors or other small molecules.

The structures of GVL1, TAT, TSHα, and Amphotericin B underwent molecular docking using the Molegro Virtual Docker software, version 6.0.1 (MVD) [67]. The molecular geometry was first optimized using molecular mechanics (MMFF94), and the most stable conformation of each compound was submitted for molecular docking. All water molecules were deleted from the enzyme structures. The enzyme and compound structures were prepared using the default parameter settings in the software package (ligand evaluation: Internal ES, Internal H-Bond, Sp2-Sp2 Torsions, all checked; number of runs: 10 runs; algorithm: MolDock SE; maximum interactions: 1500; max. population size: 50; max. steps: 300; neighbor distance factor: 1.00; max. number of poses returned: 5).

The docking procedure was performed using GRID with a 17 Å radius and 0.30 resolution to cover the ligand-binding site of each protein. The grid box was applied to the center of the target site. The MolDock scoring algorithm was used, along with the MolDock search algorithm [68]. Molegro Virtual Docker generated five poses for each molecule in the active site of each protein. The most stable pose, i.e., the one with the lowest interaction energy, was selected and imported to the LigPlot program for visual inspection [69].

## 4. Conclusions

The bioconjugate guanidine–CPP showed low micromolar IC_50_ values against *L. amazonensis* and *L. infantum* with a high selectivity index in relation to peritoneal macrophages. The action mechanism studies indicate no activity in the membrane and enzyme inhibition on cysteine protease B. This hypothesis was confirmed by in vitro and docking studies. Docking studies also suggested that other proteins, such as trypanothione reductase, could be targeted by the bioconjugate. The data confirmed that bioconjugation increased the internalization of the guanidine inside the cells and improved the solubility of these compounds. These data indicated that guanidines conjugated with cell-penetrating peptides could be a good approach for the development of antileishmanial molecules.

## Figures and Tables

**Figure 1 molecules-30-00264-f001:**
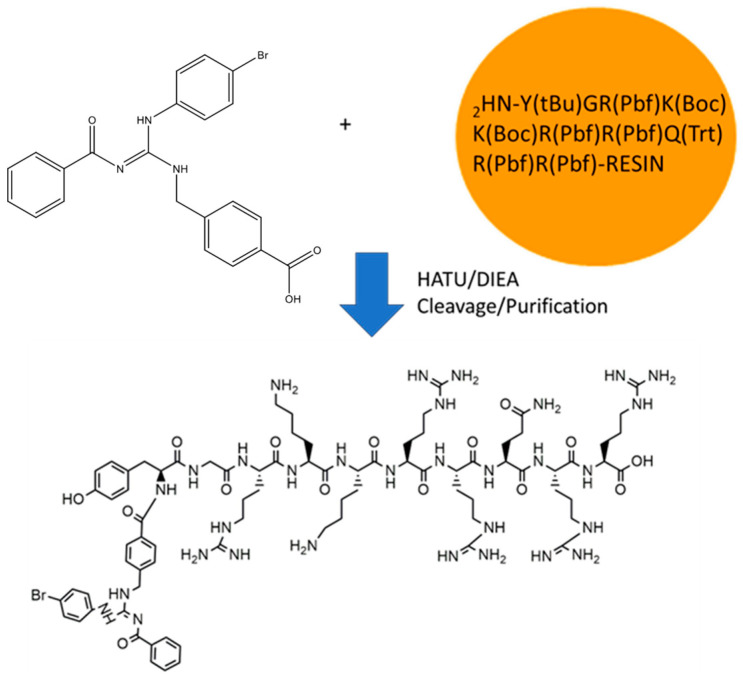
Synthesis of bioconjugate GVL1-TAT.

**Figure 2 molecules-30-00264-f002:**
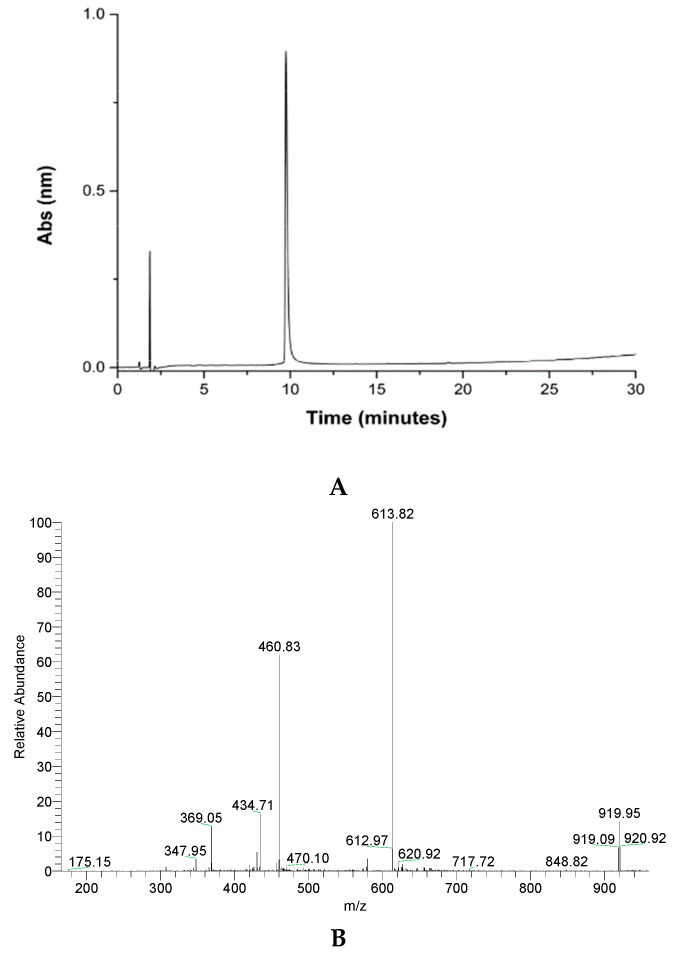
Analytical column chromatography profile of crude GVL1-TAT peptide (**A**), purified peptide (**B**), and mass spectrum of pure GVL1-TAT peptide. Theoretical MW = 1836.7 g/mol. MW/Z ratio = 919.3, 613.8, 460.2, and 368.3 for Z = 2, 3, 4, and 5, respectively.

**Figure 3 molecules-30-00264-f003:**
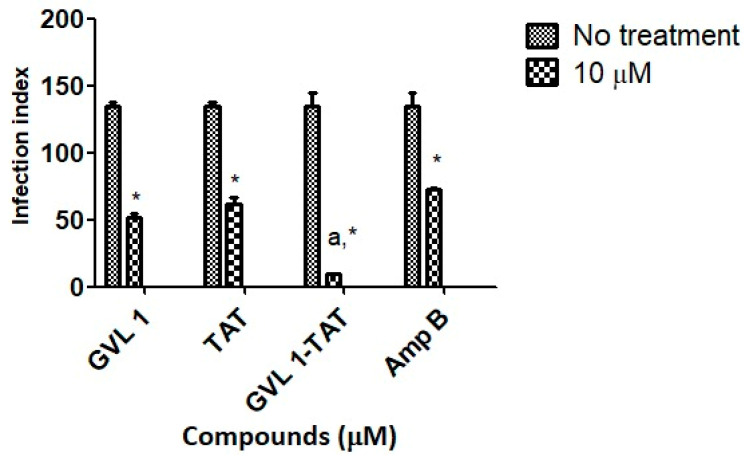
Infection indices for GVL 1, TAT, GVL 1-TAT, and Amp B in intracellular amastigotes of *L. amazonensis*. Infection index was calculated after 24 h of treatment with each compound. Negative control was untreated *L. amazonensis* intracellular amastigotes. Data are expressed as mean and standard deviation (SD) for three independent experiments. Two-way ANOVA (*p* < 0.05) was applied, where * indicates a significant difference between tested compounds and untreated control, and different Greek letters indicate significant difference between tested concentrations of each compound.

**Figure 4 molecules-30-00264-f004:**
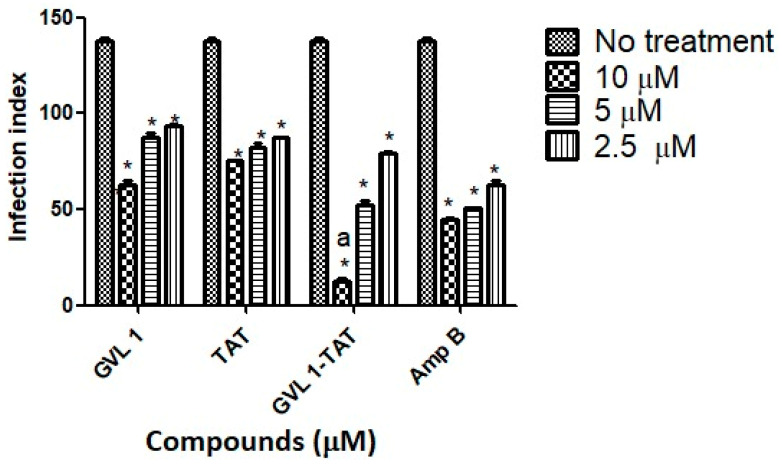
Infection indices for GVL 1, TAT, GVL 1-TAT, and Amp B in intracellular amastigotes of *L. infantum*. Infection index was calculated after 24 h of treatment with each compound. Negative control was untreated *L. infantum* intracellular amastigotes. Data are expressed as mean and standard deviation (SD) for three independent experiments. Two-way ANOVA (*p* < 0.05) was applied, where * indicates significant difference between tested compounds and untreated control, and different Greek letters indicate significant difference between tested concentrations of each compound.

**Figure 5 molecules-30-00264-f005:**
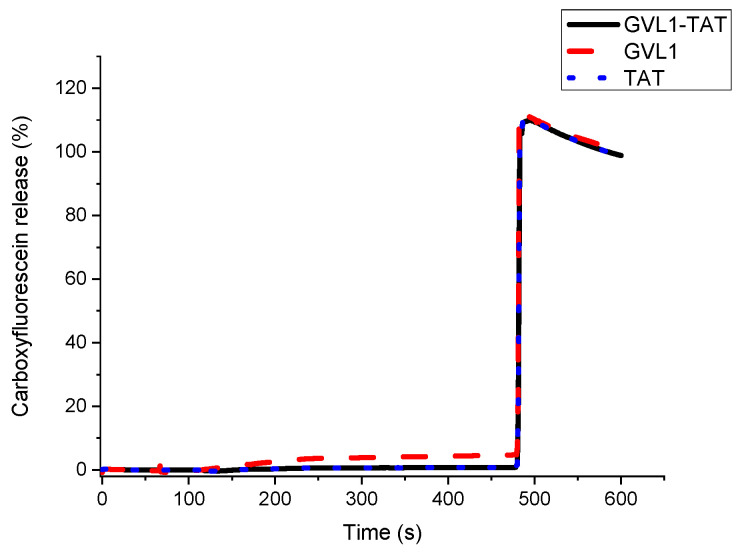
Release profile of vesicle mimicking *Leishmania* membrane containing 1-palmitoyl-2-oleoyl-sn-glycero-3-phosphocholine (POPC), 1-palmitoyl-2-oleoyl-sn-glycero-3-phospho-L-serine (POPS), and Ergosterol in presence of compounds GVL1-TAT, GVL1, and TAT at 50 μM.

**Figure 6 molecules-30-00264-f006:**
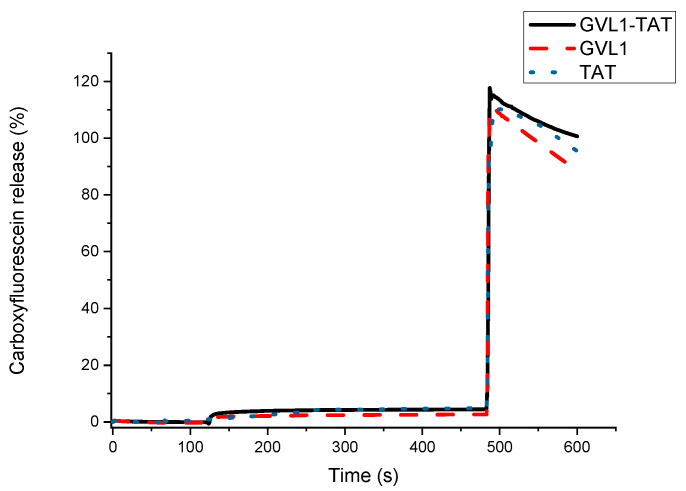
Release profile of vesicle mimicking eukaryotic membrane containing 1-palmitoyl-2-oleoyl-sn-glycero-3-phosphocholine (POPC) and cholesterol in presence of compounds GVL1-TAT, GVL1, and TAT at 50 μM.

**Table 1 molecules-30-00264-t001:** Results for antileishmanial assays with TAT, GVL1, and GVL1-TAT against *L. amazonensis*.

Compound	Promastigote IC_50_ (μM)	Peritoneal Macrophages CC_50_ (μM)	Amastigote IC_50_ (μM)	SI
TAT	NA	>1000 ± 0.15	NA	NA
GVL1-TAT	5.32 ± 0.74	>1000 ± 0.65	0.80 ± 0.06	>1250
GVL1	51.8 ± 0.10	>1000 ± 0.013	7.5 ± 3.53	>267
Amp B	3.3 ± 0.2	99.8 ± 3.7	0.75 ± 0.1	>131

NA—not active.

**Table 2 molecules-30-00264-t002:** Results for antileishmanial assays with TAT, GVL1, and GVL1-TAT against *L. infantum*.

Compound	Promastigote IC_50_ (μM)	Peritoneal Macrophages CC_50_ (μM)	Amastigote IC_50_ (μM)	SI
TAT	NA	>2000 ± 0.01	NA	NA
GVL1-TAT	1.2 ± 0.3	>2000 ± 0.65	3.8 ± 0.3	>526
GVL1	50.8 ± 2.2	>2000 ± 0.01	9.5 ± 2.5	>210
Amp B	5.4 ± 1.4	24 ± 0.1	1.25 ± 0.5	19

NA—not active.

**Table 3 molecules-30-00264-t003:** IC_50_ values (concentration capable of killing 50% of parasites) obtained in anti-promastigote assays with *L. amazonensis* and *L. infantum* in active serum.

Compounds	Promastigote IC_50_ (µM) *L. amazonensis*	Promastigote IC_50_ (µM) *L. infantum*
GVL1	65 ± 2.4	50 ± 1.2
TAT	NA	NA
GVL1-TAT	2.6 ± 0.8	1.2 ± 0.3
Amp B	0.5 ± 0.1	0.3 ± 0.1

NA—not active.

**Table 4 molecules-30-00264-t004:** Results for CPB enzyme inhibition.

Compound	CPB Inhibitory Activity 20 μM (%)	IC_50_ (μmol L^−1^)
TAT	9.42 ± 0.01	-
GVL1-TAT	91.97 ± 1.70	3.78 ± 0.1
GVL1	13.98 ± 2.8	-

**Table 5 molecules-30-00264-t005:** Summary of parameters corresponding to results obtained in molecular docking.

Compounds	Cysteine Protease	Trypanothione Reductase
GVL1	−98.136	−151.151
GVL1-TAT	−173.934	−188.683
TAT	−136.364	−179.726
Amp B	−195.256	−242.512

**Table 6 molecules-30-00264-t006:** Interactions of TAT, GVL1, and GVL1-TAT bioconjugate with residues of cysteine protease active sites from *Leishmania*. Catalytic site residues are bold.

Molecules	Hydrogens Bonds	Hydrophobic Interactions
GVL1	-	Ala 100, Lys 101, Leu 104, Asn 105, Pro 106, Asp 107, Tyr 108, Ser 111, Glu 189, Asn 192
TAT	-	Leu 149, Gln 179, Asn 192, Tyr 217, Thr 220, Gly 223, Gly 224, Thr 225, Pro 227, His 230
GVL1-TAT	Ala 100, Lys 101, Leu 102, Tyr 103, Leu 104, Asn 105, Asp 107, Tyr 108, Lys 145, Gln 147, Gly 148, Leu 149, Cys 150, **Cys 153**, Phe 156, Gly 160, Glu 189, Cys 191, Asn 192, Asn 288, **His 289**, Gly 290, Val 291	Lys 90, Phe 91, Leu 94, Pro 106, Tyr 109, Ser 111, His 112, Pro 143, Val 144, **Gly 151**, Ser 157, Ile 159, Asn 161, Glu 178, Gly 193, Gly 194, Val 262, Ala 263, Val 264, Leu 292, Lys 308, **Asn 309**, Trp 311, Gly 312
Amp B ^1^	Lys 101, Leu 104, Asn 105, Pro 106, Asp 107, tyr 109, Thr 110, **Gly 151**, Cys 191, Asn 192, Gly 194, Thr 269, Ser 286, Asn 288	Pro 96, Ala 100, Leu 102, Tyr 103, Tyr 108, Leu 149, Gly 193, Leu 287, **His 289**

^1^ Positive control.

## Data Availability

The original contributions presented in this study are included in the article/Appendix A.

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
