# Peer review of "Guanidines Conjugated with Cell-Penetrating Peptides: A New Approach for the Development of Antileishmanial Molecules"

_molecules, 2025, doi:10.3390/molecules30020264_

Round 1
Reviewer 1 Report
Comments and Suggestions for Authors
The manuscript entitled “Guanidines conjugated with cell-penetrating peptides: A new approach for the development of antileishmanial molecules” is focused on a well-designed study exploring the bioconjugation of guanidine compounds with cell-penetrating peptides (CPPs) as a promising strategy to develop effective antileishmanial agents. The research shows significant results, including enhanced leishmanicidal activity of the developed GVL1-TAT bioconjugate against Leishmania amazonensis and Leishmania infantum, with high selectivity indices and lower toxicity than individual compounds. Enzyme inhibition assays further elucidated the mechanism of action of the bioconjugates while docking studies confirmed the higher interaction of the conjugated with CPP and suggested that other proteins. Additionally, the docking analysis suggested that other crucial parasitic proteins, such as trypanothione reductase, may also serve as viable targets for these bioconjugates. These findings highlight the versatility of GVL1-TAT in disrupting key enzymatic pathways within Leishmania spp., paving the way for future investigations into its broader spectrum of activity.
I will recommend the acceptance after major revisions (listed below).
1. To improve the contextualisation of the manuscript on development-based approaches to guanidine compounds as potential treatments against parasite diseases, authors should refer to works exploring guanidine derivatives as anti-parasitic agents. Relevant articles include:
DOI 10.3390/ph15111341, DOI 10.1021/acs.jnatprod.6b00256, and DOI: 10.1016/j.ejmech.2024.116162
These papers emphasise the importance of guanidine-containing molecules in targeting different parasite proteins and/or metabolic pathways and could provide further insight into their therapeutic potential.
2. Considering that the three-dimensional structures of the trypanothione reductase enzyme and cysteine protease proteins were obtained using the AlphaFold web tool, authors could include a paragraph discussing the critical role of crystallographic structures of parasitic proteins in drug discovery and molecular modelling. High-resolution X-ray crystallographic data are essential for understanding enzyme mechanisms, active site architecture, and protein-ligand interactions. These structural insights provide the foundation for accurate docking studies and the rational design of effective inhibitors. In cases where experimental crystal structures are unavailable, alternative approaches such as homology modelling and the use of advanced tools like AlphaFold become invaluable. Homology modelling relies on structurally similar templates to build reliable models of the target protein, while AlphaFold offers a powerful method for predicting protein structures de novo. The addition of this paragraph would enhance the manuscript’s methodological transparency and strengthen the interpretation of the docking results.
Suggested reference: DOI 10.1016/bs.armc.2018.08.005 studies emphasizing the role of crystallographic data in studying parasitic proteins and DOI: 10.3390/molecules25051030, DOI: 10.3389/fcimb.2022.944748.
3. The authors should place more emphasis on the significant finding that the GVL1-TAT conjugate can potentially target two distinct proteins. This dual-targeting capability is a valuable finding that highlights the therapeutic potential of the conjugate. To emphasise the importance of this result, the authors could refer to studies in which similar dual-targeting strategies have been implemented with promising results. For instance, articles such as
DOI: 10.3390/ph14070636, DOI: 10.3390/molecules28227526, and DOI: 10.1096/fj.201901342R
provide excellent examples of the efficacy and relevance of this approach in targeting parasite enzymes, as this strategy represents a robust and innovative solution to address the challenges of resistance and efficacy of parasite treatments.
Minor Suggestions
Graphic representation:
The addition of comparative tables or diagrams summarising IC₅₀ values, selectivity indices and binding energies would improve the clarity and accessibility of the results.
Typographical and grammatical revisions:
Correct minor typographical errors
Author Response
Reviewer 1.
- To improve the contextualisation of the manuscript on development-based approaches to guanidine compounds as potential treatments against parasite diseases, authors should refer to works exploring guanidine derivatives as anti-parasitic agents. Relevant articles include:
DOI 10.3390/ph15111341, DOI 10.1021/acs.jnatprod.6b00256, and DOI: 10.1016/j.ejmech.2024.116162
Thank you for your comments and suggestions. It was done.
- Considering that the three-dimensional structures of the trypanothione reductase enzyme and cysteine protease proteins were obtained using the AlphaFold web tool, authors could include a paragraph discussing the critical role of crystallographic structures of parasitic proteins in drug discovery and molecular modelling. High-resolution X-ray crystallographic data are essential for understanding enzyme mechanisms, active site architecture, and protein-ligand interactions. These structural insights provide the foundation for accurate docking studies and the rational design of effective inhibitors. In cases where experimental crystal structures are unavailable, alternative approaches such as homology modelling and the use of advanced tools like AlphaFold become invaluable. Homology modelling relies on structurally similar templates to build reliable models of the target protein, while AlphaFold offers a powerful method for predicting protein structures de novo. The addition of this paragraph would enhance the manuscript’s methodological transparency and strengthen the interpretation of the docking results.
Suggested reference: DOI 10.1016/bs.armc.2018.08.005 studies emphasizing the role of crystallographic data in studying parasitic proteins and DOI: 10.3390/molecules25051030, DOI: 10.3389/fcimb.2022.944748.
Thank you for your comments and suggestions. It was done.
- The authors should place more emphasis on the significant finding that the GVL1-TAT conjugate can potentially target two distinct proteins. This dual-targeting capability is a valuable finding that highlights the therapeutic potential of the conjugate. To emphasise the importance of this result, the authors could refer to studies in which similar dual-targeting strategies have been implemented with promising results. For instance, articles such as
DOI: 10.3390/ph14070636, DOI: 10.3390/molecules28227526, and DOI: 10.1096/fj.201901342R
provide excellent examples of the efficacy and relevance of this approach in targeting parasite enzymes, as this strategy represents a robust and innovative solution to address the challenges of resistance and efficacy of parasite treatments.
Thank you for your comments and suggestions. It was done.
Graphic representation:
The addition of comparative tables or diagrams summarising IC₅₀ values, selectivity indices and binding energies would improve the clarity and accessibility of the results.
In our opinion, this would result in duplication of data in the manuscript text. The data can be seen in the text.
Typographical and grammatical revisions:
Correct minor typographical errors
It was done. The English certification was added.
Reviewer 2 Report
Comments and Suggestions for Authors
The work here depicted clearly describes the potential use of Guanidines conjugated with cell-penetrating peptides, in this case TAT, against Leishmania. I believe that this work can be very interesting for the scientific community since it demonstrates that, by using peptides conjugated with small molecules, it is possible to considerably increase the molecules' biological potential.
Furthermore, this is a very well structured manuscript, starting by the evaluation against the two stages of the parasite and following through permeability and stability assays, and finally studies against potential molecular targets, achieving a quite complete evaluation of this particular conjugate.
Thus, I truly believe in the value of this work and support its publication.
Nevertheless, I would like to suggest some improvements:
1) I would remove Figure 7 (or at least place it as SI) since it is considerably confusing and it presents the same information present on Table 6.
2) A "Conclusion" section is missing in the manuscript. I believe that a section were the compilation of the main conclusions provided by the results and discussion would be very beneficial for summarizing the interesting insights developed in this work.
Author Response
The work here depicted clearly describes the potential use of Guanidines conjugated with cell-penetrating peptides, in this case TAT, against Leishmania. I believe that this work can be very interesting for the scientific community since it demonstrates that, by using peptides conjugated with small molecules, it is possible to considerably increase the molecules' biological potential.
Furthermore, this is a very well structured manuscript, starting by the evaluation against the two stages of the parasite and following through permeability and stability assays, and finally studies against potential molecular targets, achieving a quite complete evaluation of this particular conjugate.
Thus, I truly believe in the value of this work and support its publication.
Thank you for your comments and suggestions.
Nevertheless, I would like to suggest some improvements:
1) I would remove Figure 7 (or at least place it as SI) since it is considerably confusing and it presents the same information present on Table 6.
2) A "Conclusion" section is missing in the manuscript. I believe that a section were the compilation of the main conclusions provided by the results and discussion would be very beneficial for summarizing the interesting insights developed in this work.
It is done.
Reviewer 3 Report
Comments and Suggestions for Authors
A new approach for the development of antileishmanial molecules explores an innovative strategy for developing new antileishmanial agents by conjugating the cell-penetrating peptide (CPP) TAT with the guanidine compound GVL1. The study introduces a novel approach by combining CPPs with guanidines, which holds potential to overcome limitations of current antileishmanial therapies, such as resistance and toxicity.
The findings demonstrate that the GVL1-TAT conjugate exhibits high selectivity and enhanced antileishmanial activity compared to individual compounds. Molecular docking studies further support the proposed mechanism of action and suggest additional protein targets, adding depth to the analysis.
While docking studies suggest potential interactions with trypanothione reductase and other proteins, experimental validation of these targets is necessary to confirm their roles in the observed activity.
The study mentions the potential to reduce resistance and toxicity but does not address potential challenges in clinical application, such as drug delivery, stability, or safety in vivo.
Comments on the Quality of English Language
no
Author Response
New approach for the development of antileishmanial molecules explores an innovative strategy for developing new antileishmanial agents by conjugating the cell-penetrating peptide (CPP) TAT with the guanidine compound GVL1. The study introduces a novel approach by combining CPPs with guanidines, which holds potential to overcome limitations of current antileishmanial therapies, such as resistance and toxicity.
The findings demonstrate that the GVL1-TAT conjugate exhibits high selectivity and enhanced antileishmanial activity compared to individual compounds. Molecular docking studies further support the proposed mechanism of action and suggest additional protein targets, adding depth to the analysis.
Thank you for your comments and suggestions.
While docking studies suggest potential interactions with trypanothione reductase and other proteins, experimental validation of these targets is necessary to confirm their roles in the observed activity.
The study mentions the potential to reduce resistance and toxicity but does not address potential challenges in clinical application, such as drug delivery, stability, or safety in vivo.
Thanks for your suggestions. The experimental validation using trypanothione reductase and clinical application will be done in another manuscript.
Round 2
Reviewer 3 Report
Comments and Suggestions for Authors
I have reviewed this revised manuscript. The authors have well improved the manuscript and have properly addressed my concerns. So, this work can be recommended now to be accepted for publication in Molecules.